# Bio-Actives from Natural Products with Potential Cardioprotective Properties: Isolation, Identification, and Pharmacological Actions of Apigenin, Quercetin, and Silibinin

**DOI:** 10.3390/molecules28052387

**Published:** 2023-03-05

**Authors:** Ekaterina-Michaela Tomou, Paraskevi Papakyriakopoulou, Helen Skaltsa, Georgia Valsami, Nikolaos P. E. Kadoglou

**Affiliations:** 1Department of Pharmacy, School of Health Sciences, National and Kapodistrian University of Athens, Panepistimiopolis, 15784 Athens, Greece; 2Medical School, University of Cyprus, Nicosia 2029, Cyprus

**Keywords:** apigenin, quercetin, silibinin, silymarin, isolation, identification, cardiovascular disease, diabetes mellitus, hypertension, metabolic syndrome

## Abstract

Cardiovascular diseases (CVDs) are the leading cause of morbidity and mortality worldwide. As a result, pharmaceutical and non-pharmaceutical interventions modifying risk factors for CVDs are a top priority of scientific research. Non-pharmaceutical therapeutical approaches, including herbal supplements, have gained growing interest from researchers as part of the therapeutic strategies for primary or secondary prevention of CVDs. Several experimental studies have supported the potential effects of apigenin, quercetin, and silibinin as beneficial supplements in cohorts at risk of CVDs. Accordingly, this comprehensive review focused critically on the cardioprotective effects/mechanisms of the abovementioned three bio-active compounds from natural products. For this purpose, we have included in vitro, preclinical, and clinical studies associated with atherosclerosis and a wide variety of cardiovascular risk factors (hypertension, diabetes, dyslipidemia, obesity, cardiac injury, and metabolic syndrome). In addition, we attempted to summarize and categorize the laboratory methods for their isolation and identification from plant extracts. This review unveiled many uncertainties which are still unexplored, such as the extrapolation of experimental results to clinical practice, mainly due to the small clinical studies, heterogeneous doses, divergent constituents, and the absence of pharmacodynamic/pharmacokinetic analyses.

## 1. Introduction

Cardiovascular diseases (CVDs) are a cluster of diseases involving disorders in the structure and/or functionality of cardiac tissue as well as vascular integrity [1]. Cardiovascular mortality and morbidity are still among the leading global health issues, as it is responsible for more than 17 million deaths each year according to the World Health Organization [2]. In this context, bio-actives of natural products have become especially meaningful as part of the prevention and management of CVDs. Lifestyle interventions (e.g., healthy diets) have been strongly incorporated into international health policies. Particularly, plant-based and Mediterranean diets are recommended as the most health-beneficial diets [3,4], as they include diverse components with a broad range of nutrients and bioactive phytochemicals such as flavonoids.

The latter represents one of the largest groups of metabolites with diverse chemical structures [5]. Over time, numerous flavonoids have been isolated/identified and extensively investigated for their pharmacological activities, especially for their potential to decrease the risk of CVDs [6,7,8]. The cardioprotective effects of different flavonoids include antihypertensive, vasorelaxant, anti-atherosclerotic, and antithrombotic activities [6,7]. Flavonoids are categorized into different classes such as flavones, 3-hydroxyflavones (flavonols), flavans, etc. Among them, it is estimated that flavones (e.g., apigenin) and flavonols (e.g., quercetin) are greatly associated with lower coronary heart disease mortality [9] and CVD occurrence [10,11,12,13].

Flavonolignans are phenols of mixed origin; a part of flavonoid and of phenylpropane. They were first discovered in milk thistle [*Silybum marianum* (L.) Gaertn.] with silibinin being the most studied and bioactive flavonolignan [14,15]. Over the years, the hepatoprotective activity of silymarin extract and its major constituent, silibinin, has become widely known [16], and recent studies have shown their cardioprotective effects as well [15,17,18].

This review aims to summarize the isolation and identification methods of apigenin, quercetin, and silibinin from natural products to be assessed as potential cardioprotective agents. Their cardioprotective properties were critically explored and commented based on in vitro, preclinical, and clinical studies associated with atherosclerosis and a wide range of cardiovascular risk factors including hypertension, diabetes, dyslipidemia, obesity, cardiac injury, and metabolic syndrome (MS).

## 2. Literature Search Strategy 

An extensive search was performed in the electronic databases PubMed, Scopus, EMBASE, and MEDLINE for English-language publications from 2000 to December 2022. The article’s research was focused on isolation/identification studies of each bio-active: apigenin, quercetin, and silibinin. The search was based on the following terms: isolated apigenin/quercetin/silymarin/silibinin/silybinin/milk thistle and cardiovascular diseases/hypertension/diabetes/dyslipidemia/atherosclerosis/obesity/cardiac injury/metabolic syndrome Moreover, experimental and clinical studies (meta-analyses of randomized clinical trials and cohort studies) from the literature were screened and selected using the terms: apigenin, quercetin, silymarin, silibinin, silybinin, milk thistle and atherosclerosis, diabetes, hyperglycemia, hypertension, hyperlipidemia, metabolic syndrome, coronary artery disease, reperfusion/ischemia, cardiac injury. Two investigators (E-M.T. and P.P.) performed the literature search independently excluding the studies with full text unavailable, publication language other than English, conference abstracts, and studies based on the mixing of each of the bio-actives with other substances. 

Specific names (i.e., genus, species, authority, and family) of the plants are quoted in the main text as reported in the related publication. However, their names and families based on the database “IPNI” [19] were mentioned inside parentheses in Table 1 at the end of Section 3.1.

Using the abovementioned terms, we initially found 3961 hits. After the screening of titles and abstracts, we removed 3750 irrelevant studies; 211 full-text studies were assessed for eligibility. After removing the studies with irrelevant outcomes and unavailable full text, we ended up with a total of 61 isolation/identification studies, 106 experimental studies, 20 systematic reviews and meta-analyses of clinical data, 6 additional clinical studies which were not included in those meta-analyses, and 1 cohort study. The increasing number of studies included in this review are chronologically presented in Figure 1.

## 3. Results and Discussion

### 3.1. Chemical Structure, Plant Origin (Family), Methods of Isolation, and Identification

In this section, the chemical structures (Figure 2) and the methods of isolation/identification of each bio-active (i.e., apigenin, quercetin, and silibinin) from natural sources related to cardioprotective studies are thoroughly described and summarized in Table 1 at the end of this section. 

#### 3.1.1. Apigenin

Apigenin (4′,5,7-trihydroxyflavone) is one of the most universal flavonoids in vegetables, fruits, and herbs. Its name is obtained from the genus *Apium* (Apiaceae family). Some main sources of apigenin are parsley, celery, and chamomile [10,20]. The chemical structure has hydroxyl groups at positions C-5 and C-7 of the A-ring and at C-4′ of the B-ring and belongs to a class of flavonoids known as flavones (Figure 2). It is worth mentioning that the bioactivity of apigenin is related to its chemical structure (indicating a structure–activity relationship) since specific groups (e.g., a hydroxyl group or double bond) could be associated with a specific biological activity [10]. For instance, Paredes et al. (2018) mentioned that the presence of a double bond among the positions C-2 and C-3 of the C-ring and of the 4′-hydroxyl group at the B-ring in an aglycon form could play an important role in provoking vascular relaxation and improving eNOS expression [21]. Many studies focusing on the cardioprotective properties have reported the method for the isolation and identification of this bioactive compound from different plant extracts.

##### Methods for Isolation: Column Chromatography and Preparative HPLC

A methanol extract (70%) and the major constituents of *Ailanthus excelsa* Roxb. leaves (Simaroubaceae family) were studied for their in vitro hypotensive activities by Loizzo et al. (2007) [22]. The bioactive ethyl acetate fraction was subjected to column chromatography over silica gel with increasing polarity system solvent CHCl_3_:MeOH. Apigenin was obtained after further fractionations with column chromatography over Sephadex LH-20. The major flavonoids (apigenin and rutin) from the methanol extract of *Teucrium polium* L. (Lamiaceae family) were investigated for their potential to trigger insulin secretion in an STZ-induced diabetes model (rat pancreatic islets) and their mechanism of action [23]. Regarding the isolation, the methanol extract from the aerial plant parts was subjected to column chromatography over silica gel with different solvents, and then apigenin was purified from a fraction after column chromatography over Sephadex LH-20 with methanol. The same methodology was used by Esmaeili and Sadeghi (2009), who also reported in vitro antiglycation activity of these flavonoids isolated from the specific plant [24]. Apigenin was isolated from the ethyl acetate fraction of the ethanol extract of *Platycodon grandiflorum* (Jacq.) A. DC. (Campanulaceae family), among other constituents, by using column chromatography over Sephadex LH-20 with CH_2_Cl_2_:MeOH as the eluent [25]. All the obtained compounds were evaluated for the in vitro inhibitory effects on the formation of advanced glycation end products (AGEs) and rat lens aldose reductase (RLAR). Apigenin did not exhibit important activities compared to the other studied compounds and the positive control. Interestingly, they reported that flavones with two hydroxyl groups at position C-3′ and C-4′ at the B-ring (such as luteolin) demonstrated greater inhibitory activity in both assays than flavones with one hydroxyl group at the B-ring (i.e., apigenin). Chaves et al. (2011) isolated apigenin using column chromatography over Sephadex LH-20 (EtOH) from the ethyl acetate fraction of the aqueous extract of *Petroselinum crispum* (Mill.) Nym. ex A.W. Hill (common name parsley; Apiaceae family), and then they evaluated its anti-platelet activity [26]. In another study, apigenin was isolated from the chloroform extract from the leaves of *Premna foetida* Renw. ex Blume (Lamiaceae family) [27]. It was also identified in the methanol extract (19.0 µg/mL) by the RP-HPLC method using a gradient system of H_2_O:ACN. Senejoux et al. (2012) investigated *Ziziphora clinopodioides* Lam. (Lamiaceae family) and its constituents through a bioassay-guided strategy to determine their vasodilating properties using an in vitro model of rat isolated thoracic aortic rings [28]. The hydroalcoholic extract showed vasorelaxant activity (Emax 50.3 ± 4.9%), while the chloroform fraction obtained from the hydroalcoholic extract was the most active, as previously reported [29]. Subsequently, this fraction was further studied and subjected to column chromatography on Sephadex LH-20 with the solvent system CH_2_Cl_2_:MeOH (1:1), which yielded six fractions. Among these fractions, fractions D and E demonstrated great vasorelaxant effects and were further fractionated with flash column chromatography over silica gel. Apigenin was isolated and identified from fraction E. All the isolated compounds, categorized into flavones, cinnamic acid derivatives, and acetophenones, were evaluated for their vasorelaxant effects. Furthermore, the 70% methanol extract from the leaves of *Matricaria recutita* L. (common name German chamomile; Asteraceae family) and its constituents were investigated for their inhibitory effects on rat lens aldose reductase (RLAR), advanced glycation end products (AGEs), and antioxidant activity [30]. The extract was subjected to column chromatography over Sephadex LH-20 (MeOH), and it gave 16 fractions. Apigenin was isolated by further column chromatography over Sephadex LH-20 of these fractions. Dou et al. (2020) isolated apigenin as the major flavonoid of the 70% methanol extract from the flowers of *Gentiana veitchiorum* Hemsl. (Gentianaceae family) by using column chromatography over silica gel with CHCl_3_:MeOH (100:1 to 1:1) [31]. To obtain apigenin, the fraction eluted by 3:1 CHCl_3_:MeOH was further purified by semi-preparative HPLC (MeCN:H_2_O). The extracts and the obtained compounds were evaluated for their inhibition of ACE. Apigenin was isolated by preparative HPLC from the 80% methanol extract of *Morus indica* L. (Moraceae family) and was studied for its hypoglycemic potential in streptozotocin (STZ)-induced diabetic rats [32]. In another study, apigenin was isolated from the ethyl acetate extract fraction (75% ethanol) of *Sophora alopecuroides* L. (Leguminosae family) along with many flavonoids by using column chromatography on Sephadex LH-20 (MeOH) [33]. In this study, the isolated compounds were explored for their potential to improve insulin resistance in an in vitro insulin resistance cell model. Vo Van et al. (2022) studied the flavonoid content and the in vitro/in vivo antidiabetic activity of the extract of *Merremia tridentata* (L.) Hallier f. (Convolvulaceae family) [34]. Apigenin was identified and quantified by HPLC-DAD analysis with four other flavonoids, and it was isolated from the extract by column chromatography over silica gel. An in silico study was also carried out to predict their binding affinity with antidiabetic receptors.

##### Methods for Identification: HPLC and LC-MS Analysis

Rossoni et al. (2005) investigated the aqueous extract and its most abundant flavonoids, namely apigenin and luteolin, from the wild artichoke (*Cynara cardunculus* L.-Asteraceae family) for their vasorelaxant activities on isolated rat aortic rings [35]. Furthermore, they studied whether the vasomodulating effects could be observed after daily administration of the aqueous extract (10 mg/kg/day of polyphenols) by oral gavage in aged rats for five days. Apigenin and luteolin were identified using the HPLC method. The extracts and two main flavonoids (i.e., apigenin and luteolin) from the powdered flowers of *Chrysanthemum morifolium* Ramat. (Asteraceae family) were studied for their effects on oxLDL-induced expression of ICAM-A and E-selectin in human aortic and umbilical vein cells (HUVEC) as well as on HL-60 cell adhesion [36]. Apigenin and luteolin were identified by an LC-MS technique in the hot aqueous (HCM; 309 and 1149 µg/g dry weight, respectively) and the ethanol (ECM; 2110 and 1689 µg/g dry weight, respectively) extracts. Dou et al. (2020) identified apigenin as the major flavonoid of the 70% methanol extract of *Gentiana veitchiorum* Hemsl. (Gentianaceae family) using an HPLC-MS/MS technique [31]. Apart from the antioxidant activity, apigenin was studied for its antihyperlipidemic effects.

#### 3.1.2. Quercetin

Quercetin (3,3′,4′,5,7-pentahydroxyflavone) is a ubiquitous flavonoid present in almost all edible vegetables and fruits as well as in most plants. Its name originates from the Latin “quercetum” (oak forest, quercus oak) [37]. Some main sources of quercetin are onions, asparagus, red-leaf lettuce, apples, berries, cherries, red wine, and tea infusions [37]. Quercetin belongs to the class of flavonoids known as 3-hydroxyflavones (flavonols) (Figure 2). The chemical structure has hydroxyl groups at positions C-3 of the C-ring, C-5 and C-7 of the A-ring, and C-3′ and C-4′ of the B-ring. It is noteworthy that the noticeable bioactivities of apigenin are related to its chemical structure (indicating a structure–activity relationship), especially to the presence of functional groups such as hydroxyl groups [37,38,39]. For instance, the antioxidant activity of quercetin is related to (i) the orthodihydroxy or 3′,4′-catechol group, (ii) the 3- and 5-hydroxyl groups, and (iii) the Δ2 double bond close to a 4-oxo group [37,39]. However, many quercetin derivatives could be formed, including glycosides, methylated versions, and rarely occurring sulfate and prenyl substituents. In general, quercetin is commonly found as a derivative in glycosidic forms mainly conjugated with glucose and rutinose [37]. The most common glycosylation position is the hydroxyl group at C-3 [37,38]. A growing number of studies focusing on the cardioprotective properties have described the isolation and identification process of this bioactive compound from different plant extracts. 

##### Methods for Isolation: Column Chromatography and Preparative HPLC

Estrada et al. (2005) isolated quercetin from the methanol extract of *Bauhinia megalandra* Griseb. (Leguminosae family) by column chromatography on Sephadex LH-20 (MeOH). The obtained constituents were also evaluated for their antihyperglycemic effects in a rat liver microsomal glucose-6-phosphatase (G-6-Pase) bioassay. Quercetin was isolated from a 70% ethanol extract of *Carya illinoinensis* (Wangenh.) K. Koch (Juglandaceae family) [40]. The crude extract was subjected to vacuum liquid chromatography (VLC) to afford fractions with different polarities. One of the fractions was further fractionated by repeated column chromatography techniques, and quercetin was obtained. In addition, the antidiabetic effects of the obtained compounds were examined. Li et al. (2011) investigated the chemical constituents of *Cyclocarya paliurus* (Batal.) Iljinsk. (Cyclocaryaceae family) and their glucosidase and glycogen phosphorylase inhibition [41]. The chloroform fraction derived from 75% ethanol extract of the bark was fractionated by column chromatography on silica gel with a solvent mixture of chloroform, ethyl acetate, and methanol. Its fraction (eluted with CHCl_3_:MeOH 99:1) was further purified by column chromatography on silica gel (petroleum ether: EtOAc 100:0 to 0:100) and Sephadex LH-20 (H_2_O:MeOH 100:0 to 50:50), which yielded quercetin. 

A methanol extract from the aerial parts of *Polygonum hyrcanicum* Rech. f. (Polygonaceae family) was prepared and then partitioned [42]. Its ethyl acetate residue was subjected to normal phase column chromatography to produce seven fractions. Further purification by column chromatography on a Sephadex LH-20 column eluted with MeOH:EtOAc (2:1) led to the isolation of quercetin, which was evaluated for its in vitro α-glucosidase inhibitory capacity. Quercetin was isolated from the methanol extract of *Artemisia capillaris* Thunb. (Asteraceae family) and tested for its activity against α-glucosidase and protein tyrosine phosphatase 1B (PTP1B) [43]. The 50% ethanol extract from the leaves of *Allium victorialis* L. (Liliaceae family) was partitioned, the ethyl acetate residue was further purified by column chromatography techniques, and quercetin was obtained [44]. Quercetin was derived after continuous fractionations by column chromatography of an ethyl acetate fraction from the methanol extract of *Dillenia indica* Blanco (Dilleniaceae family) [45]. All the obtained constituents were evaluated for their antidiabetic activity. Quercetin was isolated from the ethanol extract of the flower buds of *Coreopsis tinctoria* Nutt. (Asteraceae family) and was evaluated for its vasorelaxant activity [46]. Specifically, the ethanol extract was applied to AB-8 resin, which produced six eluates. Among them, two fractions (CTAD and CTAE) were rich in flavonoids. The CTAD fraction was fractionated by the ODS RP-18 column, and further purifications by chromatography on Sephadex LH-20 (MeOH) led to the isolation of quercetin. Moreover, the 75% ethanol extract from *Sarcopyramis nepalensis* Wall. (Melastomataceae family) was studied [47]. After applying the extract to liquid–liquid extraction, the ethyl acetate was subjected to an AB-8 macroporous adsorption resin column with different polarity systems of ethanol, and quercetin was obtained by column chromatography over Sephadex LH-20 (MeOH). The isolated compounds were tested for their in vitro α-glucosidase inhibitory activity.

The ethanol extract of *Cuscuta pedicellata* Ledeb. (Convolvulaceae family) was investigated for its activity on body weight and serum lipid profile in the high-fat diet (HFD)-fed rats animal model [48]. A bio-guided strategy was applied, and ten compounds were isolated from the bioactive fractions. Among them, quercetin was isolated from the ethyl acetate fraction after repeated purification by column chromatography techniques such as flash silica chromatography and Sephadex LH-20. Quercetin obtained from the ethanol extract of *Toona sinensis* (A. Juss.) M. Roem. (Meliaceae family) was evaluated for its antihyperglycemic activity [49]. Specifically, the 70% ethanol extract from the leaves was partitioned, the ethyl residue was fractionated by column chromatography (n-hexane: EtOAc:MeOH), and quercetin was obtained by capillary electrophoresis using silica gel column chromatography. Furthermore, an ethanol extract and its fractions from the stems of *Tetracera indica* Merr. (Dilleniaceae family) were examined for their antidiabetic potential [50]. Quercetin was isolated after continuous fractionations by column chromatography over silica gel and Sephadex LH-20 of an ethyl acetate fraction. Owis et al. (2017) investigated the effects of the constituents of *Cordia boissieri* A. DC. (Boraginaceae family) on MS (in vivo study) [51]. The hydroalcoholic extract from leaves was partitioned, and the ethyl acetate residue was subjected to column chromatography on polyamide. The obtained fractions were further fractionated, and quercetin was isolated from one of the subfractions after using column chromatography over Sephadex LH-20. In another study, quercetin was isolated by repeatedly column chromatography techniques from the ethyl acetate residue of the hydroalcoholic extract of *Xenophyllum poposum* (Phil.) V. A. Funk (Asteraceae family) [52]. It was also identified in the crude extract by the HPLC-DAD-MS/MS method using a gradient system of 1% formic acid in water and methanol. They also explored the vasodilating and hypotensive potential of the extract and its isolated compounds.

The methanol extract from the leaves of *Bryophyllum pinnatum* (Lam.) Oken (Crassulaceae family) was fractionated, and its bioactive ethyl acetate residue was further analyzed by column chromatography on silica gel with an isocratic solvent system MeOH:EtOAc:H_2_O (5:3:2) [53]. Quercetin was isolated from one of the obtained fractions. The isolated compounds were explored for their antidiabetic activities. In addition, quercetin was isolated from the methanol extract of *Phyllanthus emblica* L. (Euphorbiaceae family) by column chromatography over silica gel with chloroform and methanol as solvent mixtures (increasing polarity; methanol up to 90%) [54]. In this study, the antihyperglycemic activity of quercetin was evaluated in vivo in streptozotocin (STZ)-induced diabetic rats. Moreover, quercetin was obtained from the methanol extract of *Lactuca serriola* L. (Asteraceae family) and tested for its in vitro α-glucosidase inhibitory activity [55]. Fadul et al. (2020) investigated the antiglycation effects of isolated compounds from *Geigeria alata* (DC), Oliv. and Hiern. (Asteraceae family) [56]. Quercetin was obtained by several chromatography columns on silica gel with DCM:MeOH as the solvent system. The methanol extract from the leaves of *Coreopsis lanceolata* L. (Asteraceae family) was prepared and then partitioned by liquid–liquid extraction [57]. Its ethyl acetate fraction was subjected to reverse-phase column chromatography (RP-CC) with a gradient solvent system MeOH:H_2_O (1:1 to 1:0 *v*/*v*), which produced nine fractions (F01-F09). Fraction F07 was further fractionated by RP-CC (CH_3_CN:H_2_O 4:6 to 1:0 *v*/*v*) and then purified by column chromatography on Sephadex LH-20 with MeOH (100%) to yield quercetin. All the isolated compounds were evaluated for their antidiabetic effects. Furthermore, quercetin was isolated using column chromatography on Sephadex LH-20 from the ethyl acetate residue of the methanol extract from *Cynanchum acutum* L. (Apocynaceae family) [58]. Quercetin was also isolated using preparative RP-HPLC from the hot water extract of *Acacia arabica* (Lam.) Willd. (Leguminosae family) [59].

The 70% ethanol extract and its isolated compounds from the leaves of *Crataegus pinnatifida* Bge. var. *major* N.E.Br. (Rosaceae family) were studied for their in vitro effects on lipid metabolism [60]. Quercetin was isolated using column chromatography methods over silica gel and Sephadex LH-20. In another study, quercetin was obtained from the ethyl acetate fraction of an extract (75% ethanol) of *Sophora alopecuroides* L. (Leguminosae family), along with many flavonoids, using column chromatography on Sephadex LH-20 (MeOH) [33]. Quercetin was also isolated through a bio-guided approach to antidiabetic activity from the ethanol extract of the stems of *Bauhinia strychnifolia* Craib (Leguminosae family) [61]. The extract was subjected to liquid–liquid extraction, and then the bioactive ethyl acetate fraction was further purified by column chromatography over silica gel and Sephadex LH-20. The bioactive compounds were also identified through LC-QTOF/MS analysis in this fraction. Zhang et al. (2022) explored the antidiabetic effects of the constituents of *Pueraria thomsonii* Benth. (Fabaceae family) [62]. Its ethanol extract from the leaves was further extracted with different solvents, and the ethyl acetate fraction was selected. Using column chromatography over silica gel and SHP-20P, quercetin was isolated. In addition, the ethyl acetate fraction was analyzed via the HPLC-DAD method.

##### Methods for Identification: HPLC and LC-MS Analysis

Bernatoniene et al. (2014) identified quercetin from the 70% ethanol extract of the aerial parts of *Leonurus cardiaca* L. (Lamiaceae) by using the HPLC method [63]. The aqueous extract of the fruits from *Ugni molinae* Turcz. (Myrtaceae) was investigated for its cardiovascular potential [64]. Its constituents, which included quercetin, were identified by the HPLC technique with a gradient solvent system of 1% formic acid: acetonitrile. Moreover, quercetin was identified in the ethyl acetate fraction of the hydroalcoholic extract from the leaves of *Mandevilla moricandiana* Woodson (Apocynaceae family) [65]. The extract was subjected to liquid–liquid extraction to give five fractions. The ethyl acetate fraction was fractionated by column chromatography on Sephadex LH-20, which yielded seven subfractions (MMEAF-A to MMEAF-G). Quercetin was identified by UHPLC-DAD-ESI-MSn analysis in MMEAF-F. The ethanol extract and its major compounds from *Anacardium humile* A.St.-Hil. (Anacardiaceae family) were studied for their antidiabetic effects [66]. Quercetin was identified using HPLC-ESI-MS/MS analysis.

#### 3.1.3. Silymarin Extract and Constituents

Silymarin is a phenolic mixture extracted from *Silybum marianum* (L.) Gaertn. (Asteraceae family) fruits, commonly known as milk thistle. It is reported that silymarin was first isolated in 1968 by Wagner et al. [16,67]. Generally, silymarin is composed of seven flavonolignans (about 80%; silybin A, silybin B, isosilybin A, isosilybin B, silychristin, isosilychristin, and silydianin) and the flavonoid taxifolin [68,69]. Flavonolignans are a combination of flavonoid and lignan structures [70]. In particular, they are formed by oxidative coupling reactions among a flavonoid and a phenylpropanoid, frequently coniferyl alcohol. Among them, silibinin (also called silibinine, silybine, silybin, etc.) is the principal bioactive component of silymarin and comprises 50–70% of it [15,67]. Silibinin’s structure consists of a taxifolin and coniferyl alcohol unit and is a mixture of the two diastereoisomers, silybins A and B (Figure 2) [68,69,70]. Isosilybin, a regioisomer, is also a mixture of diastereoisomers, isosilybins A and B [68,69,70]. It is important to mention that several factors could influence the content of flavonolignans in the silymarin mixture. This includes environmental conditions and geographical regions of origin [67]. Although the isolation of flavonolignans from silymarin is a regular process, the separation of the pure bioactive diastereoisomers is yet a challenging procedure since these compounds are so similar [71]. To date, silibinin is still considered by many authors as a single chemical entity, resulting in many misconceptions. However, some examples of the reported analytical methods include high-performance liquid chromatography (HPLC) coupled to ultraviolet (UV), diode-array detector (DAD), mass spectrometry (MS), tandem mass spectrometry, and column-switching HPLC with electrochemical detections [16].

##### Methods for Isolation and Identification

Studies focused on the isolation or identification of silymarin flavonolignans and their cardioprotective activities are limited. Silymarin constituents were identified through HPLC-DAD, which reported six flavonolignans [silybin A (12.7%), silybin B (21.7%), isosilybin A (4.5%), isosilybin B (21.7%), silychristin (16.1%), and silydianin (7.1%)) and one flavonoid (taxifolin (2.6%)] [69]. Furthermore, silychristin was isolated by using preparative HPLC. Palomino et al. (2016) investigated the ethyl acetate extract from the seeds of *S. marianum* using the HPLC method with solvent system phosphoric acid:methanol:water in gradient elution [72]. They identified silychristin, silydianin, silibinin A, silibinin B, isosilibinin A, and isosilibinin B in the extract. In addition, the ethanol:water (1:1) extract of *S. marianum* was analyzed by HPLC-DAD, which revealed the following flavonoids and flavonolignans: taxifolin, silychristin, apigenin-7-glucoside, silydianin, silybin A, silybin B, isosilybin A, and isosilybin B [73].

**Table 1 molecules-28-02387-t001:** Methods of isolation and identification of apigenin, quercetin, and silibinin from plants.

Botanical Name (Family)	Extract/Residue-Fraction	Plant Parts	Method/Solvents	References
**Apigenin**				
*Ailanthus excelsa* Roxb. [*A. excelsus* Roxb.] (Simaroubaceae)	70% Methanol/Ethyl acetate (isolation)	L.	CC (Sephadex LH-20)	[22]
*Chrysanthemum morifolium* Ramat. (Asteraceae)	Aqueous, Ethanol (identification)	Fl.	LC-MS	[36]
*Cynara cardunculus* L. (Asteraceae)	Aqueous (identification)	L.	HPLC analysis	[35]
*Gentiana veitchiorum* Hemsl. (Gentianaceae)	70% Methanol (identification)	Fl.	HPLC-MS/MS 0.1% formic acid/water and methanol	[31]
	70% Methanol (isolation)		CC (silica gel)/CHCl_3_-MeOH (100:1 to 1:1), Semi-prep HPLC/MeCN-H_2_O	
*Matricaria recutita* L. (Asteraceae)	70% Methanol (isolation)	L.	CC (Sephadex LH-20)/acetone	[30]
*Merremia tridentata* (L.) Hallier f. (Convolvulaceae)	Aqueous, 50% Ethanol (isolation; identification)	Stem; R.	CC (silica gel)/MeOH, CHCl_3_	[34]
			HPLC-DAD	
*Petroselinum crispum* (Mill.) Nym. ex A.W. Hill (Apiaceae)	Aqueous/Ethyl acetate (isolation)	L.	CC (Sephadex LH-20)/EtOH	[26]
*Premna foetida* Renw. ex Blume (Lamiaceae)	Methanol (identification)	L.	RP-HPLC/0.1% H_3_PO_4_: ACN (gradient system)	[27]
	Chloroform (isolation)		CC	
*Platycodon grandiflorum* (Jacq.) A. DC. [*P. grandiflorum* A. DC.] (Campanulaceae)	Ethanol/Ethyl acetate (isolation)	Fl.	CC (silica gel)/CH_2_Cl_2_: MeOH (19:1 to 9:1)	[25]
*Morus indica* L. (Moraceae)	80% Methanol (isolation)	L.	prep-HPLC	[32]
*Sophora alopecuroides* L. (Leguminosae)	75% Ethanol/Ethyl acetate (isolation)	A.p.; R.; S.	CC (Sephadex LH-20)/MeOH	[33]
*Teucrium polium* L. (Lamiaceae)	Methanol (isolation)	A.p.	CC (silica gel)/different solvent systems	[23,24]
			CC (Sephadex LH-20)/MeOH	
*Ziziphora clinopodioides* Lam. (Lamiaceae)	Hydroalcoholic (80% Ethanol:20% Water)/Dichloromethane (isolation)	Whole plant	CC (Sephadex LH-20) Flash CC (silica gel)	[28]
**Quercetin**				
*Acacia arabica* (Lam.) Willd. (Leguminosae)	Hot water (isolation)	B.	RP-HPLC	[59]
*Allium victorialis* L. (Alliaceae; Liliaceae ^p^)	50% Ethanol/Ethyl acetate (isolation)	L.	CC	[44]
*Anacardium humile* A.St.-Hil. (Anacardiaceae)	98% Ethanol (identification)	L.	HPLC-ESI-MS/MS)/water acidified with formic acid (0.1% *v*/*v*) and MeOH	[66]
*Artemisia capillaris* Thunb. (Asteraceae)	Methanol (isolation)	Whole plant	CC	[43]
*Bauhinia megalandra* Griseb. (Leguminosae)	Methanol/Ethyl acetate-acetone (8:2) (isolation)	L.	CC (Sephadex LH-20)	[74]
*Bauhinia strychnifolia* Craib. (Leguminosae)	Ethanol/Ethyl acetate (isolation; identification)	Stem	CC (Sephadex LH-20)	[61]
			LC-QTOF/MS	
*Bryophyllum pinnatum* (Lam.) Oken (Crassulaceae)	Methanol/Ethyl acetate (isolation)	L.	CC (silica gel)/ MeOH:EtOAc:H_2_O (5:3:2)	[53]
*Carya illinoinensis* (Wangenh.) K. Koch (Juglandaceae)	70% Ethanol (isolation)	B.	CC	[40]
*Cordia boissieri* A.DC. (Boraginaceae)	Hydroalcoholic/Ethyl acetate (isolation)	L.	CC (polyamide, Sephadex LH-20)	[51]
*Coreopsis lanceolata* L. (Asteraceae)	Methanol/Ethyl acetate (isolation)	Fl.	RP-CC/MeOH:H_2_O; CH_3_CN:H_2_O, CC (Sephadex LH-20)/MeOH	[57]
*Coreopsis tinctoria* Nutt. (Asteraceae)	Ethanol (isolation)	Flower buds	ODS-RP-18 column/MeOH: H_2_O, CC (Sephadex LH-20)/MeOH	[46]
*Crataegus pinnatifida* Bge. var. *major* N.E.Br. [*C. pinnatifida* f. *major* (N.E.Br.) W.Lee] (Rosaceae)	70% Ethanol	L.	CC	[60]
*Cuscuta pedicellata* Ledeb. (Convolvulaceae)	Ethanol (isolation)	Whole plant	CC	[48]
*Cyclocarya paliurus* (Batal.) Iljinsk. (Juglandaceae; Cyclocaryaceae ^p^)	75% Ethanol/Chloroform (isolation)	B.	CC (silica gel, Sephadex LH-20)	[41]
*Cynanchum acutum* L. (Asclepiadaceae; Apocynaceae ^p^)	Methanol/Ethyl acetate (isolation)	Whole plant	CC (Sephadex LH-20)	[58]
*Dillenia indica* Blanco (Dilleniaceae)	Methanol/Ethyl acetate (isolation)	L.	CC	[45]
*Geigeria alata* (DC), Oliv. and Hiern. [*G. alata* Benth. and Hook.f. ex Oliv.] (Asteraceae)	80% Ethanol/Chloroform, Ethyl acetate (isolation)	n.d.	CC (silica gel)/DCM:MeOH	[56]
*Lactuca serriola* L. (Asteraceae)	Methanol (isolation)	A.p.	n.d.	[55]
*Leonurus cardiaca* L. (Lamiaceae)	70% Ethanol (identification)	A.p.	HPLC	[63]
*Mandevilla moricandiana* Woodson (Apocynaceae)	Hydroalcoholic (70% Ethanol: 30% Water)/Ethyl acetate (identification)	L.	UHPLC-DAD-ESI-MS^n^	[65]
*Phyllanthus emblica* L. (Euphorbiaceae)	Methanol (isolation)	Fr.	CC (silica gel)/CHCl_3_: MeOH	[54]
*Polygonum hyrcanicum* Rech.f. (Polygonaceae)	Methanol/Ethyl acetate (isolation)	A.p.	CC (silica gel, Sephadex LH-20)	[42]
*Pueraria thomsonii* Benth (Fabaceae)	75% Ethanol/Ethyl acetate (isolation; identification)	L.	CC (silica gel, SHP-20P)	[62]
			HPLC-DAD	
*Sarcopyramis nepalensis* Wall. (Melastomataceae)	70% Ethanol/Ethyl acetate (isolation)	Whole plant	CC (Sephadex LH-20/MeOH)	[47]
*Sophora alopecuroides* L. (Leguminosae)	75% Ethanol/Ethyl acetate (isolation)	A.p.; R.; S.	CC (Sephadex LH-20)/MeOH	[33]
*Tetracera indica* Merr. [*T. indica* (Christm. and Panz.) Merr.] (Dilleniaceae)	Ethanol/Ethyl acetate (isolation)	Stems	CC (Silica gel, Sephadex LH-20)	[50]
*Toona sinensis* (A.Juss.) M.Roem. (Meliaceae)	80% Ethanol/Chloroform, Ethyl acetate (isolation)	L.	CC (silica gel)/n-hexane:EtOAc:MeOH, capillary electrophoresis using silica gel CC	[49]
*Ugni molinae* Turcz. (Myrtaceae)	Aqueous (identification)	Fr.	HPLC/1% HCOOH: ACN	[64]
*Xenophyllum poposum* (Phil.) V.A.Funk (Asteraceae)	Hydroalcoholic (Ethanol:Water, 1:1)/Ethyl acetate (isolation; identification)	A.p.	CC	[52]
			HPLC-DAD-MS/MS	
**Flavonolignans and extracts of *Silybum marianum* (L.) Gaertn. (Asteraceae)**				
Silymarin constituents	n.d. (identification)	n.d.	HPLC-DAD/H_2_O + 0.1% HCOOH; MeOH + 0.1% HCOOH	[69]
Silychristin	n.d. (isolation)		prep-HPLC	[69]
*S. marianum*	Ethyl acetate (identification)	S.	HPLC/H_3_PO_4_: MeOH: H_2_O (0.5:35:65–0.5:50:50 *v*/*v*/*v*)	[72]
*S. marianum*	Ethanol:Water (1:1) (identification)	S.	HPLC-DAD/ water with 0.1% formic acid; MeOH (1:1)	[73]

A.p., aerial parts; B., bark; CC, column chromatography; Fl., flowers; Fr., fruits; L., leaves; n.d, not determined; ^p^, as referred in publication; R., roots; S., seeds.

### 3.2. Physicochemical and Biopharmaceutical Properties

#### 3.2.1. Apigenin

The molecular formula of apigenin is C_15_H_10_O_5_, and its molecular weight (MW) is 270.24 g/mol [20,75]. Purified apigenin is a yellow powder with low solubility in water, but it is soluble in organic solvents such as methanol, ethanol, and dimethyl sulfoxide (DMSO) [75]. Apigenin is characterized by poor solubility in lipophilic and highly hydrophilic solvents, while its maximum solubility is noted in phosphate buffers with pH 7.5.

Furthermore, it shows high permeability through the intestinal membrane, and, consequently, it is classified in class II according to the Biopharmaceutical Classification System (BCS) [76]. Although apigenin absorption takes place along the entire gastrointestinal tract, its overall bioavailability is considered low, a fact that limits apigenin applicability in clinical practice [75]. Apigenin is mainly excreted unabsorbed or metabolized in glycosides after absorption [76]. Its transport across biological membranes can be mediated by active carriers (in the duodenum and jejunum) or via passive diffusion (in the duodenum, jejunum, ileum, and colon) [77]. The t1/2 ranges from 1.8–4.2 h, with an average value of 2.52 ± 0.56 h [78]. Apigenin metabolism is a result of methylation, sulfation, and glucuronidation processes involving both Phase I and II enzymes [77]. However, conjugation reactions such as glucuronidation and sulfation are the main metabolic pathways forming 3-monoglucuronides and apigenin-7-sulfate [79]. Consequently, apigenin is commonly found in glycosylated forms in nature, as these derivatives are more soluble than aglycon, which is unstable; apigenin conjugation with β-glycoside improves its bioavailability [80]. Moreover, apigenin oxidation by CYP enzymes lead to chrysin and luteolin products, which also present significant cardioprotective activity [81,82]. Apigenin excretion occurs mainly in urine and feces, where the 73% of the oral given dose can be found [76].

#### 3.2.2. Quercetin

The molecular formula of quercetin is C_15_H_10_O_7_, and its molecular weight (MW) is 302.24 g/mol [37,83]. Quercetin is a polyphenolic secondary metabolite, classified in flavonoids, found in several vegetables and fruits such as broccoli, red onion, berries, grapes, tea leaves, etc. [84]. The five hydroxyl groups are responsible for the antioxidant properties of the compound, and among these, the groups mainly responsible are the highly active catechol and 4-oxo groups of the B and C ring, respectively [13]. It is a yellow-colored powder characterized by high lipophilicity and low bioavailability. 

Quercetin is considered as a class IV compound showing low aqueous solubility and low stomach absorption, followed by more extensive intestinal absorption, after oral administration [85]. It also undergoes extensive first-pass metabolism, a fact that further limits its bioavailability. However, the solubility of quercetin derivatives differs depending on the substituents. For instance, glycosylation of its hydroxyl groups increases its hydrophilicity, and it mostly occurs in the 3-OH group, resulting in the most abundant and active metabolite, quercetin-3-*O*-β-d-glucuronide [38]. It is also a substrate of sulfotransferases (SULTs), uridine-50-diphosphate glucuronosyl transferases (UGTs), and catechol-*O*-methyltransferases (COMTs) that lead to the formation of several metabolites. The absorption is mediated either via ATP-binding cassette (ABC) transporters or through the deglycosylation process in enterocytes and hepatocytes [86]. The aglycone portion, which is released, can be passively diffused into the hepatic portal vein or undergo phase I and II metabolism [85]. The absorbed fraction is bound to serum albumin, while the metabolites are transported to the liver. Quercetin metabolites can be identified in the bloodstream 30 min after the oral administration of the pure substance. A significant fraction of these metabolites is cleared from the blood in 24 h. Major quercetin metabolites have also been detected in urine, indicating the contribution of the kidneys in flavonol’s removal [86]. About 20 to 60% of quercetin metabolites are eliminated through the urine, while the rest are eliminated through the lungs and the feces [87].

#### 3.2.3. Silymarin Extract and Constituents

Silymarin is a lipophilic extract that includes several flavonoid-like compounds referred to as flavonolignans, which are not soluble in water [67]. The reference composition of this extract consists of silibinin (33.4%), silychristin (12.9%), silydianin (3.5%), and isosilybin (8.35%) [88]. Silibinin has two diastereomers, and it is the most abundant component of the extract. It is hydrophobic and characterized by low solubility in water, ethanol, and methanol solvents [14]. Interestingly, silibinin solubility in water increases precipitously with pH and slightly with temperature [14]. It is not soluble in non-polar solvents such as chloroform and petroleum ether. However, silibinin is soluble in acetone, dimethylformamide (DMF), and tetrahydrofuran (THF). It is noteworthy that strong bases or extended heating over 100 °C might cause changes in silibinin structure [14]. The hydroxyl groups, as well as the coupling of metal ions with the 3,4- and 4,5-positions, are strongly correlated with the high antioxidant properties of silibinin [88].

Silibinin absorption occurs along the entire gastrointestinal tract and is more extensive in the duodenum. However, the sparse studies that have been performed to assess its bioavailability claim that it is very low: not exceeding 1% [89]. The classification of silibinin according to the BCS is considered particularly challenging. The herb milk thistle (Legalon commercial product) has been proposed to belong to class III; however, the solubility of silibinin in different pH values (2, 4.5, and 6.8) has been reported to be significantly low [90]. Consequently, silibinin inclusion in class IV is considered more reasonable [15]. Silymarin flavonolignans mostly undergo extensive phase II metabolism in the liver, and the formed conjugated metabolites are excreted in bile. Glucuronidation is the predominant metabolic procedure resulting in four major silybin monoglucuronides: silybin-A-7-*O*-β-d-glucuronide, silybin-B-7-*O*-β-d-glucuronide, silybin-A-20-*O*-β-d-glucuronide, and silybin-B-20-*O*-β-d-glucuronide [91]. The pathway that enterocyte-derived metabolites follow until elimination in feces is not well understood. The main fraction is removed via the feces, while the percentage of silymarin/silibinin excreted in urine does not exceed 5% of the administered dose.

### 3.3. Bio-Actives’ Cardiovascular Prevention Activity Based on Preclinical and Clinical Studies

The cardioprotective role of apigenin, quercetin, and silymarin/silibinin has been extensively studied in several experimental studies, while clinical trials have been performed to assess the impact of quercetin and silymarin supplementation in patients with any of the following disorders: hypertension, diabetes, dyslipidemia, obesity, and MS, which highly predispose to CVDs. The experimental and clinical studies are summarized in Table 2 and Table 3, respectively, at the end of this section. It should be noticed that in the case of apigenin there are only preclinical data about its contribution to CVD management. The contribution of each bio-active in the prevention/management of CVDs is illustrated in Figure 3.

Key: ABCA1, ATP-binding cassette transporter; ACE, angiotensin-converting enzyme; ADMA, asymmetric dimethylarginine; AMPK, AMP-activated protein kinase; Bax, B-cell lymphoma protein 2-associated X; Bcl-2, B-cell lymphoma protein 2; BW, body weight; CK, creatine kinase; CXCR-4, C-X-C chemokine receptor type 4; DBP, diastolic blood pressure; ERK, extracellular-signal-regulated kinase; FBG, fasting blood glucose; GSH, glutathione; GSSG, glutathione disulfide; HDL, high-density lipoprotein; ICAM-1, intercellular adhesion molecule 1; IL-6, interleukin 6; JNK, c-Jun N-terminal kinase; LDH, lactate dehydrogenase; LDL, low-density lipoprotein; LOX-1, lectin-like oxidized low-density lipoprotein receptor-1; MAPK, mitogen-activated protein kinase; MCP-1, monocyte chemoattractant protein-1; MDA, malondialdehyde; NF-κB, nuclear factor kappa-light-chain-enhancer of activated B cells; PAH, polycyclic aromatic hydrocarbon; ROS, reactive oxygen species; SBP, systolic blood pressure; SDF-1, stromal cell-derived factor 1; SOD, superoxide dismutase; TC, total cholesterol; TG, triglycerides; VCAM-1, vascular cell adhesion molecule 1

#### 3.3.1. Hypertension

Many strategies have been proposed for the management of hypertension using bio-actives from natural products [92]. The antihypertensive activity of apigenin and quercetin has been ascribed to the modification of several pathways. Based on in vitro studies, they are both involved in reactive oxygen species (ROS) and oxidative stress reduction as well as in the suppression of inflammatory cytokines (IL-1β, IL-6, IL-10, TNF-α, and MCP-1) [11,93,94,95,96]. The antioxidant effect of both flavonoids can also be mediated via the reduction in lipid peroxides and the activation of the AMP-activated protein kinase (AMPK)/sirtuin 1 (SIRT1) pathway [97,98,99]. Less oxidative stress and the anti-inflammatory effects of SIRT1 suppress the related vascular damage and potentially reduces blood pressure [100,101]. Most importantly, the methanol extract of *A. excelsa* inhibited the angiotensin-converting enzyme (ACE) in a dose-dependent manner [22]. Similarly, quercetin inhibits ACE by binding the zinc molecule at the active site of the enzyme, but no other quantitative data are available [22,102]. 

Vasorelaxation constitutes a mainstay of antihypertensive therapy. Apigenin and quercetin have been found to reduce peripheral vascular resistance by increasing NO bioavailability [52,103,104,105] and by impeding calcium exchange across the cell membranes [46]. Another way to increase NO production involves the quercetin-induced suppression of autophagy [94,96]. Quercetin isolated from *U. molinae* (murtilla) fruits has shown vasodilatory effects mediated by calcium-dependent potassium channels [64]. *C. cardunculus* extract, apigenin, and luteolin have demonstrated dose-dependent vasorelaxation to norepinephrine-precontracted aortic rings, with Emax values of 78 ± 5%, 57 ± 3%, and 83 ± 5%, respectively (positive control: acetylcholine with Emax 97 ± 4%) [35]. Moreover, quercetin has been found to induce 100% vasorelaxation in aortic rings preconditioned with phenylephrine [46]. It is important to note that apigenin exhibits a higher activity (Emax 88.5 ± 1.1%) than the other flavones (apigenin>chrysin>thymonin>acacetin), with an EC50 value of 189.4 ± 12.4 µM [29]. HBZY-1, M1 cortical collecting duct (M1CCD), endothelial cells (HUVEC), isolated rat aortic rigs, as well as endothelial cells of mesenteric arteries have been used for in vitro studies testing apigenin (0.5 or 72.0 μM), quercetin (20 and 30 μM), or silibinin/silymarin (50 mg/L) [96,99,104,105,106,107,108] (Table 2). Endothelial dysfunction amelioration has potential to lower blood pressure.

The aforementioned findings were also confirmed by in vivo experiments based on the oral administration of apigenin (via gavage, food, or drinking water) in doses ranging from 1.44 to 100 mg/kg/day for 3 to 6 weeks [10,21,97]. In the case of quercetin, studies in spontaneously hypertensive rats (SHRs) showed that a dose > 7 mg/kg, for 5 to 12 weeks could achieve a significant decrease in both systolic and diastolic blood pressure (SBP and DBP, respectively) [96,109,110] (Table 2). Intraperitoneal injection of 20 mg/kg of silibinin in obese diabetic mice decreased the circulating and vascular asymmetric dimethylarginine (ADMA) levels, which restored NO bioavailability [111]. 

The clinical impact of quercetin supplementation in hypertension is disputable. It has been found that its effect is more pronounced when given in higher doses (>500 mg/day) and in hypertensive patients with MS or who smoke. Additionally, seven RCTs have documented suppression of only SBP [112]. The very recent meta-analysis of Popiolek-Kalisz and Fornal (2022), which included ten trials of pre-hypertensive and normotensive participants (*n* = 841), demonstrated that quercetin supplementation reduced the SBP in the total population, while DBP lowering was observed in prehypertensive participants [113]. Regarding hypertension prevention, the large prospective cohort study of Yao et al. (2021) did not notice any effect of low quercetin intake (24.7 ± 13.8 mg/day) on the incidence of hypertension [114].

#### 3.3.2. Diabetes

**Glucose-lowering effects:** Several cardiovascular complications are associated with diabetes mellitus (DM), such as cardiomyopathy, nephropathy, and micro-angiopathy. Apigenin, quercetin, and silymarin have been proposed as protective agents against those diabetic complications. Based on in vitro data, the three substances can improve glucose homeostasis by increasing glucose uptake via the stimulation of glucose transporter type 4 (GLUT4) translocation. They also improve insulin sensitivity. Consequently, silymarin administration reduces blood glucose levels [115,116,117,118,119], as has also been proposed for apigenin from the methanol extract of *M. indica* [32] and quercetin obtained from *P. emblica* fruit [54]. Moreover, the apigenin and rutin from the methanol extract of *T. polium*, as well as pure quercetin, were found to stimulate insulin secretion and sensitization in a diabetic animal model [24,120,121]. Silymarin can decrease the PI3K levels and the Akt phosphorylation of pancreatic cells implicating glucose-lowering properties [122,123]. Additional antidiabetic properties of apigenin from *C. morifolium* and *S. alopecuroides* extracts were determined in cell lines. Those experiments showed its ability to rescue cells from high-glucose-induced damage [104], possibly via the insulin cascade pathway [33]. Moreover, quercetin isolated from the *C. lanceolata* flower has shown in vitro dipeptidyl peptidase IV (DPP-IV) inhibitory activity with a potential glucose-lowering effect [57].

**Cardiac protection:** Τhe administration of apigenin to H9c2 cardiomyocytes and diabetic rat models (for 2 weeks to 8 months, in doses ranging from 5 to 100 mg/kg/day), showed significant antidiabetic activity and protective effects against cardiac remodeling [124,125,126]. The protective role of silymarin on diabetic cardiomyopathy is strongly supported by the literature either in cell lines (H9c2 rat embryonic heart cells) or in animal models (diabetic mice at the dose of 100 mg/kg) [127]. Its cardioprotective ability is expressed via the inhibition of myocardial fibrosis and collagen deposition through the counter-regulation of the TGF-β1/Smad signaling pathway [128]. Furthermore, silymarin can be involved in the regulation of Tropomyosin alpha-1 chain (TPM1) and Myosin Light Chain 2 (MYL-2), which are major genes for the retention of ventricular cardiac myocyte structure and function [129,130]. 

**Anti-inflammatory and anti-oxidative effects:** The inhibitory effect of apigenin and its alcoholic extract on the NF-κB/p65 pathway, Akt phosphorylation, and cell apoptosis, as well as their ability to restore Bcl-2/Bax levels, has been proven either with in vitro or in vivo studies [36,126] (Table 2). The anti-inflammatory effect of apigenin was found to be significant in cases of renal dysfunction due to diabetic nephropathy, as indicated by the reduction of TNF-α, IL-6, collagen deposition, and glomerulosclerosis [125]. The suppression of NF-κB levels has been demonstrated in either cell cultures treated with 100 g/mL of quercetin [95] or in diabetic rats treated with *C. acutum*-isolated flavonoids (quercetin-3-O-galactoside and quercetin) [49,58]. Silymarin, along with quercetin, is also involved in the inhibition of lipid peroxidation through the reduction of malonaldehyde (MDA) levels and the enhancement of the glutathione/glutathione disulfide (GSH/GSSG) ratio, respectively [72,118,131,132]. Silymarin extract has been proposed to restore the activity of pancreatic antioxidant enzymes when given orally in rats for 20 days (200 mg/kg/day) [133,134,135]. 

**Clinical studies:** The effects of quercetin at the clinical scale in patients with DM have not been established yet [136]. Nevertheless, one small study of 15 participants receiving *Eugenia punicifolia* for 3 months revealed glucose lowering correlated with basal insulin decrease [137]. That effect was ascribed to quercetin, which is the main component of the plant. A recent review of ten clinical trials over the last two decades in diabetic and/or nonalcoholic fatty liver disease (NAFLD) patients [15] highlighted a modest effect of silymarin on lowering blood glucose levels as well as lipid levels (TC, TG, and LDL) and proinflammatory cytokines (TNF-α, IL-1b, and IL-6). The given doses of silymarin ranged from 75 to 600 mg once per day or 140, 150, and 200 mg three times per day for 14 to 720 days. The glycemic profile improvement of silymarin has been also confirmed in a systematic review and meta-analysis of five RCTs including diabetic patients who received treatment for 45 days to 6 months in daily doses ranging between 200 and 600 mg [138] (Table 3). Despite the extensive description of the antidiabetic effects of silymarin, the different doses as well as the heterogeneity of the population sample make it difficult to interpret data and draw conclusions (Table 3).

#### 3.3.3. Dyslipidemia

The effect of apigenin on lipid accumulation has been studied in HepG2 cells treated with 25 μM of apigenin. This concentration was found to mitigate in vitro lipid accumulation [139]. In hyperlipidemic mice treated intragastrically with apigenin (doses ranging from 10 to 100 mg/kg/day) for 6 weeks, a significant decline in body weight, visceral fat, total cholesterol (TC), triglycerides (TG), and low-density lipoprotein (LDL) levels were observed. Quercetin administration in diabetic mice (orally, 0.025% *w/w*, for 9 weeks) and in rats (doses of 50 and 72.27 mg/kg/day for 12 and 2 weeks, respectively), showed significantly lower levels of blood lipids (TC, TG, and LDL) [53,140,141] (Table 2). Those effects were accompanied by the restoration of high-density lipoprotein cholesterol (HDL-c) levels as well as LOX-1, Bcl-2, and Bax expression [139,142,143]. The treatment of HFD mice or hyperlipidemic rats with 400 mg/kg/day of silymarin for 6 weeks or 300 to 600 mg/kg of silibinin for 60 days, respectively, resulted in a decrease in TC, TG, LDL, and VLDL levels as well as an increase in HDL concentrations [144,145,146].

The clinical effect of quercetin on lipid levels has been assessed in the meta-analyses of Sahebkar et al. (2017) and Tabrizi et al. (2020), which reached different conclusions [147,148]. Particularly, doses higher than 50 mg/day were reported as beneficial for TG reduction in participants with different backgrounds (healthy, type 2 diabetes, obesity, and hypertension) [147], whereas in patients with MS, TC, LDL, and TG levels were unaffected by quercetin supplementation. On the other hand, doses higher than 150 mg/day managed to lower only the LDL levels in overweight and obese individuals [149,150]. Thus, the impact of silymarin on the lipid profile is considered questionable, as the results of the different meta-analyses are not in accordance, particularly due to the high heterogeneity of the tested human population sample. Specifically, the studies assessed by Voroneanu et al., 2016 [138] did not show any significant effect of silymarin on lipid levels, whereas four other larger meta-analyses in patients with DM and hyperlipidemia reported amelioration of LDL, HDL, insulin, and MDA levels. In those studies, the received doses ranged from 280 to 2100 mg/day, and the treatment duration was extended from 45 days to 12 months [151,152,153] (Table 3).

#### 3.3.4. Atherosclerosis 

Most of the in vitro studies to assess the effect of apigenin and quercetin on atherosclerosis are performed in macrophage or endothelium cell lines, as they are considered leading actors of atherogenesis. Apigenin, in concentrations ranging from 25 to 50 μM, can reduce macrophage viability and stimulate apoptosis and autophagy [154], while it can inhibit foam cell formation at the concentration of 2 μM [155]. Treatment with 200 μM of quercetin may also inhibit foam cell formation in vitro (human THP-1-derived macrophage cells) [156]. The same findings were reported by Cui et al. (2017) after an 8-week oral treatment with 12.5 mg/kg/day of quercetin in ApoE−/− mice [157]. It is important to note that 15 and 25 μM of quercetin prevents the overexpression of the ICAM-1 and MCP-1 genes and, consequently, cell migration in atherogenic plaques [156,158]. In the context of anti-inflammatory effects, apigenin, silymarin, and quercetin can increase the expression of ABCA1 and ABCA1-mediated cholesterol efflux and limit the inflammation regulating the TLR-4/NF-κB signaling pathway at both mRNA and protein levels [159,160,161,162,163,164]. From a mechanistic perspective, the atheroprotective activity of silibinin/silymarin is ascribed to its antioxidant properties, especially towards the oxidation of LDL, as well as suppressed atherogenic inflammation. This finding follows from in vitro experiments in cells treated with 20 and 10 μM of silymarin [165]. In ApoE−/− mice, apigenin administration (doses from 10 to 100 mg/kg/day) for 6 or 8 weeks, via an intragastric gavage method or in drinking water, confirmed the reduction of pro-inflammatory cytokines, which agreed with the later study of Gao et al. (2021) [11] in SHRs [153,154,158]. In parallel, a 4-week supplement of an HFD mouse model with 100 μg/day of quercetin showed reduced potency for atherosclerotic plaque formation [166], while it enhanced the plaque stability by impeding elastin degradation, macrophage infiltration, and MMP-9 and VCAM-1 expression [167,168,169]. Similarly, administration of 100 to 200 mg/kg/day of silymarin extract reduced atherosclerotic plaque burden in hypercholesterolemic rabbits [17] (Table 2).

#### 3.3.5. Obesity

Apigenin has shown protective activity against factors predisposing to obesity in both in vitro and in vivo studies. Those studies are summarized in the very recent review of Xu et al. (2022) [10]. In vitro studies have used N-29-2, SH-SY5Y, and 3T3-L1 mouse cells as well as human mesenchymal stem cells (hMSCs) [170,171,172]. In those studies, apigenin induced AMPK phosphorylation and reduced the fatty-acid-binding protein 4 and stearoyl-CoA desaturase [173]. Apigenin can limit the expression of enzymatic activity of pancreatic lipase and fatty acid synthase [174]. According to Zhao et al. (2022), apigenin is included among six flavonoids that can ameliorate the obese conditions and insulin resistance in HFD-mice [175]. Moreover, it activates lipolysis-related genes, reducing the body weight of animals [176].

Similar anti-obesity findings regarding the quercetin flavonoid have been reported by Jung et al. (2012) [140] in C57BL/6 mice after 9 weeks of treatment with (0.025% *w*/*w* quercetin). 

Quercetin, as a CYP2E1 inhibitor, can neutralize free radicals and downregulate the mitogen-activated protein kinase (MAPK) signaling pathway, which may contribute to weight loss. Moreover, quercetin treatment of 3T3-L1 fibroblast cells limited adipogenesis through a mechanism relying on downregulation of extracellular-signal-regulated kinase (ERK) and c-Jun N-terminal kinase (JNK) phosphorylation [177]. Moreover, it acted as a weak partial agonist of peroxisome proliferator-activated receptor γ (PPARγ) and improved glucose uptake [59] more efficiently than rosiglitazone [178] (Table 2). 

The influence of quercetin supplementation on patients’ body weight is considered controversial, as several RCTs have shown its effectiveness [179,180], whereas a recent meta-analysis rejects any significant impact on body weight, body mass index (BMI), waist circumference, or waist-to-hip ratio [181] (Table 3). The variable response of patients toward quercetin treatment can be ascribed to the genetic polymorphisms of ApoE isoforms, as supported by the studies of Pfeuffer et al. (2013) and Egert et al. (2009) [149,180].

#### 3.3.6. Cardiac Injury

The cardioprotective effect of apigenin and quercetin has been tested in several in vitro and in vivo studies aiming to assess the ability of flavonoids to attenuate myocardial ischemia/reperfusion (I/R) injury. In particular, the aqueous extract of *P. crispum*, apigenin, as well as its 7-*O*-glucoside (cosmosiin), have been found to suppress platelet aggregation, with IC50 values of 1.81 mg/mL, 0.0036 mg/mL, and 0.18 mg/mL, respectively [26]. Moreover, 20 μM of quercetin in vitro and 10 mg/kg in vivo can mediate, via the PI3K/Akt and SIRT1/TMBIM6 pathways, the inhibition of cardiomyocyte apoptosis [12,182,183] and mitophagy events [184], respectively. Its antiapoptotic effect is also reflected in the low levels of apoptotic stress markers (pMAPKAPK 2, p-SAPK/JNK, p-Hsp27, p-c-JUN, and cl-CASP3) [185]. Moreover, quercetin impedes Ca^2+^ influx via L-type Ca^2+^ channels at a concentration equal to 50 μM, which protects they myocardium from I/R injury in H9C2 cells [186]. Recently, Rodrigo et al. (2022) summarized the main mechanisms involved in the cardioprotective effect of quercetin against I/R injury [187]. Using an isolated rat heart model and mouse cardiomyocyte injury model, apigenin inhibited MAPK phosphorylation and the release of MDA, lactate dehydrogenase isoenzyme (LDH), and creatine kinase (CK) cardiac markers, which are associated with heart damage and lipid peroxidation [187]. Moreover, quercetin has been reported to contribute to the inhibition of free radical generation in mitochondria as a component of the *L. cardiaca* extract [63]. Apigenin can also rescue cardiomyocytes from apoptosis by downregulating the expression of the proapoptotic factor Bax and miR-103-1-5p, which is associated with myocardial I/R injury and heart failure [188]. These findings have also been confirmed in animal studies (Table 2). The studies were performed in rats exposed in I/R injury and treated intravenously with 5 to 40 mg/kg/day of apigenin [189,190]. In the case of ischemia, pre-treatment with both flavonoids, as well as silibinin, reduced the myocardial infract size in parallel to increase in SOD activity [189,191,192]. Similarly, in mice undergoing left anterior descending artery (LAD) ligation, the intraperitoneal 7-day pretreatment of animals with 100 mg/kg of silibinin reduced ER and oxidative stress and reversed inflammation via the NF-kB pathway [193]. Finally, 50 mg/kg of quercetin given orally in rats for 30 days could reverse cardiac remodeling through the restriction of TGF-b1/Smad3 signaling [192], and the reduction of p-MAPK protein expression was observed in two different doses of quercetin (75 mg/kg and 150 mg/kg) given orally in rats [194].

#### 3.3.7. Metabolic Syndrome 

Two recent review articles aimed to outline the most important studies on the potential protective effect of apigenin against MS. Based on recently published studies about apigenin [75,77], several pathways have been proposed for its protective activity. Specifically, the nuclear factor erythroid 2-related factor 2 (Nrf2) forms a complex with the cytoplasmic Kelch-like ECH-associated protein 1 (Keap1), whose dissociation is responsible for the expression of antioxidant activity. Apigenin has been found in both in vitro and in vivo experiments, as well as in computational studies, to promote cleavage of the complex, thus enhancing the transcription of antioxidant genes. Furthermore, apigenin and silibinin can increase in vivo the NAD+ levels in the liver, thus improving lipid metabolism, glucose homeostasis, and glucose tolerance [75,195]. Both apigenin and silymarin (doses from 30 to 300 mg/kg) reduce total oxidative stress as well as the activity of gluconeogenesis regulatory enzymes and ROS production in β pancreatic cells [75,196,197]. It is worth mentioning that apigenin doses higher than 30 mg/kg can possibly cause hepatocytotoxicity events, sedation, and muscle relaxation [77], while the quercetin dose of 15 mg/kg has been found to possess hepatoprotective activity [198]. NAFLD is a common characteristic of MS. Gao et al. (2021) have studied quercetin’s effect in a NAFLD rat model and reported its protective activity [199]. Additionally, a four-herb formula proposed by Wat et al. (2018) [200] (including, among others, silymarin) showed promising results in obese C57BL/6 mice with hyperlipidemia and NAFLD, as it reduced the plasma and liver lipid content. More precisely, the milk thistle (silymarin) managed to significantly limit oleic-acid-induced fatty acid uptake into HepG2 cells [200]. Moreover, several mechanisms have been proposed to explain the effect of flavonoids on MS risk factors, such as the induction of Akt phosphorylation and glycogen synthase kinase 3 (GSK-3) as well as the downregulation of PPARα, sterol regulatory element-binding protein (SREBP) 1c, and SREBP [201]. Most of the studies in the literature have been performed in STZ-induced diabetic rats and mice in doses varying from 5 mg/kg/day to 240 mg/kg/day for a period of 10 to 90 days. 

From a clinical perspective, the ability of quercetin to reduce systolic and diastolic pressure as well as to improve HDL, TG, LDL, TC, and glucose levels has been reported in doses ranging from 30 to 1000 mg/day when the treatment lasts longer than 8 weeks [181,202]. The amelioration of those parameters has been reported in MS patients aged 60+ after the daily consumption of 240 mg for 3 months. In these cases, quercetin was also found to significantly decrease the body weight of the participants [179]. Longer quercetin treatment (≥8 weeks) in doses higher than 500 mg/day has been found to effectively derwith MS [203] (Table 3).

**Table 2 molecules-28-02387-t002:** Published preclinical studies based on the cardioprotective properties of apigenin, quercetin, and silibinin.

Cardiovascular Disease	Mechanism	Bio-Active	References
Hypertension	↓SBP and DBP	quercetin	[96,109,110]
	↓ADMA	silibinin	[18]
	↓CXCR4 and SDF-1		
	↓PAH		
	↓ROS, ↓oxidative stress, and ↓MCP-1	apigenin, quercetin	[11,93,94,95,96]
	↓overproduction of eNOS and cNOS		[21,103,104,105]
	↓lipid peroxides		[97]
	Activation of AMPK/SIRT1		[98,99]
	ACE inhibition		[22,102]
	Inhibition of calcium exchange		[46,64]
	↑NO production/bioavailability and vasorelaxation	quercetin, silibinin	[29,35,41,94,95,96]
	↓inflammatory cytokines (IL-1β, IL-6, IL-10, TNF-α, and MCP-1)	apigenin, quercetin, silibinin	[11,18,95]
Diabetes	Restoration of Bcl-2/Bax levels	apigenin	[126]
	↓ TNF-α and IL-6		[125]
	↓ CK-MB and LDH		[124]
	↑insulin release and sensitivity		[24]
	Inhibition of PKCβII activation		[104]
	↓ICAM-1 and E-selectin		[36]
	↓ROS oxidative stress	quercetin	[118]
	↑glucose uptake via GLUT4 stimulation		[115,116,117,119]
	DPP-IV inhibition		[57]
	↓ TNF-α, IL-1β, and IFNγ	silibinin/silymarin	[122,123,133]
	↓pancreatic protein damage and creatinine levels		[135]
	↓blood glucose levels	apigenin, quercetin	[32,54]
	Inhibition of myocardial fibrosis and cardiac remodeling	apigenin, silymarin	[122,126,128,129,130]
	Inhibition of lipid peroxidation, ↓MDA, and ↑GSH/GSSG ratio	quercetin, silymarin	[72,118,131,132]
	↓NF-κB/p65 and Akt phosphorylation	apigenin, quercetin, silibinin	[36,49,58,95,123,126]
Dyslipidemia	Restoration of HDL, LOX-1, and Bcl-2/Bax levels	quercetin	[139,142,143]
	↓ICAM-1, ↓IL-6, and ↓VCAM-1		[144,145,146]
	↓lipid accumulation	apigenin, quercetin	[139,146]
	↓BW		[140,141]
	↓levels of TC, TG, and LDL	apigenin, quercetin, silymarin	[140,141]
Atherosclerosis	↓proinflammatory cytokines	apigenin	[11,155,159,190]
	↓ICAM-1 and MCP-1	quercetin	[156,158]
	↓elastin degradation, ↓macrophage infiltration, and ↓MMP-9 and VCAM-1 expression		[168,169,170]
	↓LDL oxidation	silymarin	[165]
	Induction of autophagy and foam cell formation	apigenin, quercetin	[154,155,156,157]
	↓atherosclerotic plaque formation	quercetin, silymarin	[17,166]
	↑ABCA1 and ABCA1-mediated cholesterol efflux	apigenin, quercetin, silymarin	[159,160,161,162,163,164]
	↓inflammation via TLR-4/NF-κB signaling pathway		
Obesity	↓BW	apigenin, quercetin	[140,176,177]
	↑AMPK phosphorylation	apigenin	[173]
	↓fatty acid-binding protein 4 and stearoyl-CoA desaturase		
	Downregulation of MAPK, ERK, and JNK	quercetin	[177]
	↑glucose uptake	quercetin, silymarin	[59,69,160,178]
	↓fasting blood glucose levels	quercetin	[42,43,45,61,62]
	↓activity of pancreatic lipase and fatty acid synthase	apigenin	[174]
Cardiac injury	Inhibition of cardiomyocyte apoptosis via the PI3K/Akt and SIRT1/TMBIM6 pathways	quercetin	[12,182,183]
	Stimulation of mitophagy events		[184]
	Impedes Ca2+ influx via L-type Ca2+ channels		[186]
	Inhibition of MAPK phosphorylation and MDA, LDH, and CK release		[187]
	↓MAPK		[194]
	Anti-platelet activity	apigenin, quercetin	[26]
	↓LDL oxidation		[27]
	↓myocardial infract size	apigenin, quercetin, silibinin	[189,191,192]
	↑SOD activity		
	↓ER and oxidative stress, reverse of inflammation via the NF-kB pathway	silibinin	[193]
Metabolic syndrome	↑insulin secretion and sensitization	quercetin	[120,121]
	↓plasma lipid content	silymarin	[200]
	↑NAD+ levels in liver	apigenin, silymarin	[75,195]
	↓inflammatory cytokines		[75,196,197]
	↓ROS production and oxidative stress in β pancreatic cells		

ABCA1, ATP-binding cassette transporter; ACE, angiotensin-converting enzyme; ADMA, asymmetric dimethylarginine; AMPK, AMP-activated protein kinase; Bax, B-cell lymphoma protein 2-associated X; Bcl-2, B-cell lymphoma protein 2; BW, body weight; CK-MB, creatine kinase; cNOS, constitutive isoform of nitric oxide synthase; CXCR-4, C-X-C chemokine receptor type 4; DBP, diastolic blood pressure; DPP IV, dipeptidyl peptidase IV; eNOS, endothelial nitric oxide synthase; ERK, extracellular-signal-regulated kinase; GLUT-4, glucose transporter type 4; GSH, glutathione; GSSG, glutathione disulfide; HDL, high-density lipoprotein; ICAM-1, intercellular adhesion molecule 1; IFN-γ, interferon gamma; IL-10, interleukin 10; IL-1β, interleukin 1β; IL-6, interleukin 6; JNK, c-Jun N-terminal kinase; LDH, lactate dehydrogenase; LDL, low-density lipoprotein; LOX-1, lectin-like oxidized low-density lipoprotein receptor-1; MAPK, mitogen-activated protein kinase; MCP-1, monocyte chemoattractant protein-1; MDA, malondialdehyde; MMP-9, matrix metallopeptidase 9; NAD+, nicotinamide adenine dinucleotide; NF-κB, nuclear factor kappa-light-chain-enhancer of activated B cells; PAH, polycyclic aromatic hydrocarbon; PI3Ks, phosphoinositide 3-kinases; PKCβII, protein kinase C-beta II; ROS, reactive oxygen species; SBP, systolic blood pressure; SDF-1, stromal cell-derived factor 1; SIRT1, silent mating type information regulation 2 homolog) 1; SOD, superoxide dismutase; TC, total cholesterol; TG, triglycerides; TLR-4, toll-like receptor 4; TMBIM6, transmembrane BAX inhibitor motif containing 6; TNF-α, tumor necrosis factor; VCAM-1, vascular cell adhesion molecule 1; ↓, decrease; ↑, increase; ↔, non-significant change.

**Table 3 molecules-28-02387-t003:** Published reviews and meta-analyses of randomized clinical trials, clinical studies not included in meta-analyses, and a cohort study based on the cardioprotective properties of apigenin, quercetin, and silibinin.

Cardiovascular Disease	Study Design	Main Outcomes	Bio-Active	Ref.
Hypertension	Meta-analysis: Seven RCTs, 587 pts, HTN, healthy individuals	↓SBP	quercetin	[112]
	Meta-analysis: Ten RCTs, 841 pts, HTN, healthy individuals	↓SBP and DBP		[113]
	Cohort study, 15,662 pts, healthy individuals	No effect on hypertension incidence		[114]
Diabetes	Non-controlled pilot study, 15 pts, T2DM	↓glycosylated hemoglobin, ↓basal insulin, ↓TSH, ↓usCRP, ↓both SBP, ↓DBP	quercetin	[137]
	Meta-analysis: Ten clinical trials, 700 pts, healthy, T2DM, NAFLD	↓FBG, ↓HbA1c, ↓insulin, ↓TC, ↓TG, ↓LDL, ↑HDL	silymarin	[15]
	Meta-analysis: Five RCTs, 270 pts, healthy, T2DM	↓FBG, ↓HbA1c		[138]
Dyslipidemia	Meta-analysis: Five RCTs, 442 pts, healthy, T2DM, HTN, hyperlipidemia	↓TG	quercetin	[147]
	Meta-analysis: Sixteen RCTs, 1575 pts, healthy, HTN, T2DM, hypercholesterolemic	↓TC, ↔TG, ↓LDL		[148]
	Double-blinded, placebo-controlled cross-over study, 175 pts, overweight with high-CVD risk	↓LDL		[149]
	Randomized, double-blinded, placebo-controlled cross-over trial, 70 pts, overweight-to-obese patients with pre-hypertension	↔FBG, ↔LDL		[150]
	Meta-analysis: Five RCTs, 270 pts, healthy, T2DM	↔lipid levels	silymarin	[138]
	Meta-analysis: Eight RCTs, 195 pts, T2DM	↓FBG, ↓HbA1c, ↓LDL, ↓MDA, ↑HDL		[139]
	Meta-analysis: Ten RCTs, 620 pts, hyperlipidemic	↓TC, ↓TG, ↓LDL, ↑HDL		[152]
Obesity	Randomized, placebo-controlled, double-blind trial, 110 pts, MS	↓BW, ↓SBP, ↓DBP, ↓TC, ↓LDL, ↓fasting plasma insulin	quercetin	[179]
	Double-blind crossover study, 49 pts, healthy with different APOE isoforms	↓waist circumference, ↓TG, ↑HDL		[180]
	Meta-analysis: Seven RCTs, 896 pts, healthy, obese, HTN	↓SBP, ↓DBP, ↔BW, ↔BMI, ↔waist circumference, ↔waist-to-hip ratio		[181]
	Double-blinded, placebo-controlled cross-over study, 172 pts, overweight, high-CVD risk phenotype	↓SBP, ↓ox-LDL, ↔TNF-a, ↔C-reactive protein		[149]
Metabolic syndrome	Meta-analysis: Eighteen RCTs, 987 pts, HTN, overweight, MS, T2DM, NAFLD	↓SBP, ↓DBP, ↓TC, ↓TG, ↓LDL, ↑HDL, ↓glucose levels	quercetin	[202]
	Meta-analysis: Nine RCTs, 781 pts, HTN, T2DM, obesity, PCOS	↔FBG, ↔HbA1c, ↓insulin,		[203]

APOE, apolipoprotein E; BMI, body mass index; BW, body weight; CRP, C-reactive protein; CVD, cardiovascular disease; DBP, diastolic blood pressure; FBG, fasting blood glucose; HDL, high-density lipoprotein; HTN, hypertension; LDL, low-density lipoprotein; MDA, malondialdehyde; MS, metabolic syndrome; NAFLD, non-alcoholic fatty liver disease; oxLDL, oxidative modification of a low-density lipoprotein; PCOS, polycystic ovarian syndrome; pts, participants; RCTs, randomized controlled trials; SBP, systolic blood pressure; T2DM, type-2 diabetes mellitus; TC, total cholesterol; TG, triglycerides; TNF-α, tumor necrosis factor-α; TSH, thyroid-stimulating hormone; usCRP, ultra-sensitive C-reactive protein; ↓, decrease; ↑, increase; ↔, non-significant change.

## 4. Limitations of the Study

The study was based on three plant constituents (apigenin, quercetin, and silibinin) extensively assessed by in vitro and in vivo experiments as cardioprotective agents. However, there are still many uncertainties in precisely assessing the impact of such dietary supplements on humans and extrapolating experimental results to clinical practice mainly due to heterogeneous doses, divergent constituents, and the absence of pharmacodynamic/pharmacokinetic analyses. Regarding selected phytochemical studies, most of them evaluated the extracts or the isolated/identified compounds on in vitro assays without further investigation in animal or human studies. It was also observed that several of the published studies used purchased plant materials rather than isolated bio-actives. In the case of apigenin, no clinical trials have been reported to confirm the initial hypotheses of experimental assays and animal studies. Quercetin and silibinin have been clinically assessed in a wide dose range given to a small or large group of individuals with heterogeneous clinical features (healthy, hypertensive, obese, hyperlipidemic, diabetic patients, etc.). These differences among the studies explain the inconsistency of published data on several observations such as the effect of quercetin supplementation on hypertension or of silymarin on blood lipid levels.

## 5. Conclusions

The cardioprotective activity of apigenin, quercetin, and silibinin/silymarin has been demonstrated in both in vitro and in vivo experimental studies, indicating a potential contribution to lower cardiovascular morbidity and mortality. In the cases of quercetin and silymarin, limited data are also reported in clinical level, confirming their hypotensive, antidiabetic, and anti-inflammatory effects. For apigenin, no clinical trials have been reported; however, the large amount of preclinical data strongly supports its potential in CVD prevention. For the establishment of these bio-actives in clinical practice, more well-designed human trials are required. Moreover, the heterogeneity of the human population is also an issue that makes it difficult to confirm their pharmacological impact on patients.

It is important to note the need for reproducible isolation and identification methods to ensure that the isolated substances and extracts of these three bio-actives can exhibit the same actions as the purchased/synthetic materials with robustness and accuracy.

## Figures and Tables

**Figure 1 molecules-28-02387-f001:**
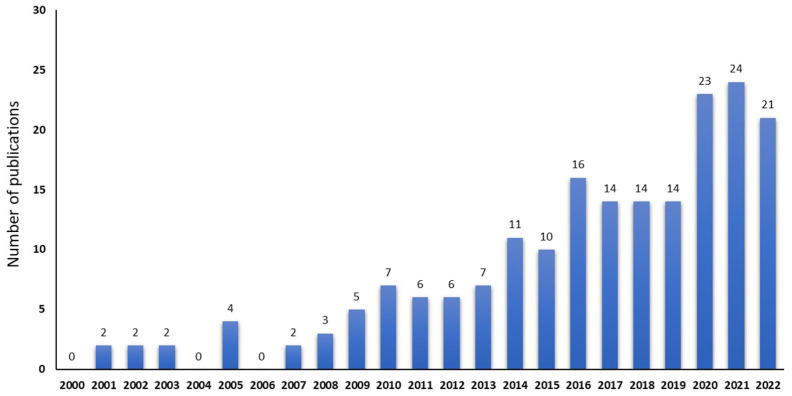
Schematic representation of the number of published studies from 2000 to 2022 based on isolation, identification, and cardioprotective activity of apigenin, quercetin, and silymarin/silibinin.

**Figure 2 molecules-28-02387-f002:**
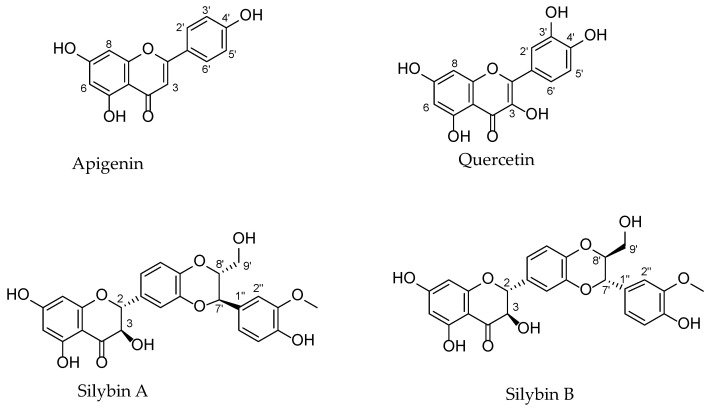
Chemical structures of apigenin, quercetin, and silybin A/silybin B (the structures were designed using the ChemDraw^®^ v.16.0 software).

**Figure 3 molecules-28-02387-f003:**
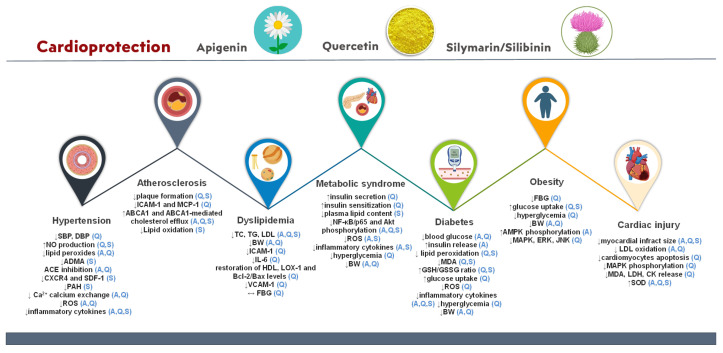
Schematic representation of the most important cardiovascular protective properties of apigenin, quercetin, and silymarin/silibinin as well as the related possible mechanisms of action. Arrows indicate increase (↑), decrease (↓), and non-significant changes (↔) in the respective biomarker expression or effect (figure animation was generated using the BioRender.com (accessed on 22 December 2022). A. apigenin; Q. quercetin; S. silibinin/silymarin.

## Data Availability

Not applicable.

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
