# Peer review of "Bio-Actives from Natural Products with Potential Cardioprotective Properties: Isolation, Identification, and Pharmacological Actions of Apigenin, Quercetin, and Silibinin"

_molecules, 2023, doi:10.3390/molecules28052387_

Round 1

Reviewer 1 Report

I read this review article with interest that aimed to provide an update on the cardioprotective properties of polyphenols, their isolation, and identification based on the available literature. The topic is of interest and highly relevant from a public health perspective due to the wide use of natural products in the general population. Much work has been done, but the submitted manuscript should be revised. I encourage the authors to consider making the necessary changes.

From a scientific point of view, the manuscript is well-organized and needs no substantial changes.

However, from a formal point of view, the number of typing errors and/or nomenclature errors should be corrected, e.g., “in vivo” and “in vitro” (lines 121, 133, 136…) in italics. Please, check the manuscript carefully, and unify the presentation.

The title is quite general and needs to reflect the work thoroughly. The title should include the names of the main active substances processed in this work.

The abstract is quite general, and I suggest the authors make significant corrections to this part of the work.

The conclusion is rather general. Make your findings more concise. In conclusion, the authors should critically review the manuscript in terms of the text's coherence and drawings.

Based on the above criticism, I recommend making minor manuscript revisions before resubmitting.

Reviewer 2 Report

I would like to commend the authors for the presentation of an excellent manuscript "Bio-actives from natural products with potential cardioprotective properties: isolation, identification, and pharmacological action"

The submitted manuscript has been thoroughly researched and provides a comprehensive overview of the subject matter. It is well written and not major errors or omissions have been noted.

While the authors have stated that commercial products containing the three listed bioactives have not been included in the scope of the research, I feel that the current availability of commercial products containing these bioactives should be mentioned in the introduction to give an indication as to whether a market exists for the materials.

As is the case with many manuscripts dealing with natural products the naming structure for each plant species does not lend itself to "easy reading" for the reader and I acknowledge the correct nomenclature has been used but I would also like to see the common name used within the sentence and that helps the reader to identify with the plant. For example on line 184 wild artichoke is named along with the botanical name and my personal view is that I prefer this style. (If the authors feel strongly about this I am happy for the common names not to be added)

Reviewer 3 Report

The review “Bio-actives from natural products with potential cardioprotective properties: isolation, identification, and pharmacological actions”, is well laid out and contains very interesting information on highly used natural products. Some observations are listed below:

1)     The authors mentioned that were excluded the studies based on purchased substances. Why? It would be interesting to include biological studies with these purchased compounds, the chemical structures are the same as those of nature. The advantage of being able to buy the flavonoids mentioned is that you can have more of them and thus more biological tests can be carried out. I don't know, valuable information could be lost in this regard. Of course, this is only an observation, and the authors can exclude from their review work what they consider pertinent.

2)     The arrangement of the information in all the tables is very confusing and looks out of order. Please review this, as the information is very interesting and important.

3)     Review figure 3, it seems that it is not very clear what the authors want to demonstrate.

4)     I recommend publishing it once they have improved the arrangement of information in the tables and have corrected some minor errors that were pointed out in the PDF file.
